# Nationalism, Post-Secular and Sufism: The Making of Neo-Bektashism by Moikom Zeqo in Post-Socialist Albania

## Gianfranco Bria

Department of Oriental Studies, University of Rome 'Sapienza', 00185 Rome, Italy; gianfranco.bria@uniroma1.it

**Abstract:** This article focuses on Moikom Zeqo's (1949–2020) work *Syri i Tretë* ("The Third Eye", 2001) as a New Age reworking of Albanian Bektashism. The success of this book, and the recognition that Bektashi authorities themselves accorded it, make it highly representative of Bektashi neo-intellectualism and beyond: it is a cross-section that enables us to investigate the complex reworking of Sufi knowledge in a post-secular environment, such as Albania. This article examines this specific work while outlining a history of the Bektashiyya from the Ottoman era to the post-socialist Albanian period and highlighting its doctrinal and practical developments. *Syri i Tretë* is the expression of a secularist engulfment of post-socialist or even post-secular religion, which Bektashism embodies. Thus, Zeqo's work expresses a common trend in Albanian society that is beyond the members of the Bektashi community.

**Keywords:** Bektashism; Albania; Sufism; Balkans; nationalism

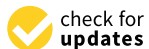



## 1. Introduction

Sufism and modernity is a somewhat debated binomial about which various scholars have written, pointing to the capacity of Sufism to both adapt and find new spaces of legitimisation and reproduction within modern societies (van Bruinessen and Howell 2007; Silverstein 2011; Weismann 2015).[1] In this paper, the focus will be on a particular *tariqa*, namely the Bektashiyya, one of the most widespread brotherhoods in the Ottoman Empire, which then gained popularity in the Balkans, especially in Albania. This popularity was mainly due to the conjunction with the nationalist movement that later led to the formation of modern Albania. This association then led, due to various circumstances that we will consider later, to the transformation from a *tariqa* to a religious community or autonomous religious sect, namely Bektashism. This development constitutes a rather interesting aspect of the present enquiry, which aims to analyse the ways in which Sufism is reasserted in modern environments. The case of Albanian Bektashism is even more singular when considering that from 1945 to 1991, the country was ruled by a Communist regime that pursued a harsh anti-religious campaign, the effects of which have been evident in the post-socialist era until the present day. Nationalism and Communism—in Albania, the latter being the sublimation of the former—are among the main driving factors of the modernisation processes, encompassing the dynamics associated with state formation. In Albania, nationalist and Communist undertones can be directly detected in Islamic practices as well as in the Islamic knowledge that has been shaped by them. This study focuses on this latter aspect and aims to analyse the evolution of Bektashi doctrines in the post-modern era as a means of investigating the modernisation of Sufism.

Specifically analysed will be the work *Syri i Tretë* ("The Third Eye") by Moikom Zeqo (1949–2020), which, in many ways, represents a New Age reworking of Albanian Bektashism. In this work, nationalist ideology is mixed with Islamic tradition and scientific knowledge in a writing style that is somewhere between the positivist and the esoteric. The latter section of this article will especially emphasise the transformations within the Albanian nation-state and society, as well as the complex impact of socialist policies and

the multifaceted post-socialist reconstruction. This article adopts a qualitative method in the critical analysis of Zeqo's text, like his other related publications. The analysis of these texts—considered primary sources—is supported by the study of Bektashi history, along with a deep knowledge of Albanian culture and society, which I have had the opportunity to study in various field research since 2014, including my doctoral research on the post-socialist Sufi revival.

## 2. Bektashi Heritage

Treated administratively as a *tariqa*, the Bektashiyya was one of the most powerful and influential Sufi orders in the Ottoman Empire (Karakaya-Stump 2022), owing to its connection with the Janissaries,[2] who had its eponymous saint, Hajji Bektash Veli (d. 1271), as their protector (Birge 1937; Faroqhi 1992; Mélikoff 1998; Yildirim 2020). The Bektashiyya acted as an accommodation room for dervish movements that were antonymous or considered disruptive to the Empire's socio-political balance, like Abdals, the forerunners of the Rumelia conquests, but also antinomian movements—such as the Melâmis-Bayrâmis and Qalandars (Antov 2017; Boykov 2021; Terzioğlu 2012). This incorporation determined the order's doctrinal and ritual corpus—influenced, moreover, by Safavid Shi'i propaganda—which has often been discussed and reflected upon, including by those who often define the Bektashiyya as a heterodox or *shari'a*-inattentive *tariqa*.[3]

The symbiotic bond with the Janissaries and Sultanic patronage allowed the Bektashi to basically bypass censure movements towards orders considered heterodox, as happened in the 17th century with the Kadizadelis.[4] In 1826, Sultan Mahmud II dissolved the order of the Janissaries and banished the Bektashis from the Empire, as blamed for being among the main factors behind the Empire's cultural, political and material backwardness (İmren Öztürk 2012; Karabulut 2017). Anyhow, the Bektashiyya experienced a particular revival among the Albanians, and various Bektashis participated in Albania's emancipation movement in different ways (Clayer 1995). Authors of Bektashi origin and background, such as the brothers Abdyl (d. 1892), Naim (d. 1900) and Sami (d. 1904) Frashëri, participated directly in the intellectual movements that laid the cultural foundations for Albanian empowerment.

Among the Frashëri brothers, Naim was certainly the most prominent in the history of the Bektashiyya. He produced intellectual works that mixed nationalism with Bektashi doctrines, many of which turned out to influence the subsequent development of the order among the Albanians. Combining pantheism, metempsychosis, nationalism and the Bektashi epic, Naim reshaped the Bektashi doctrinal framework in his *Fletore e Bektashinjët* ("Bektashi Notebook") in Bucharest in 1889. A few years later, he published the epic poem *Qerbelaja* in which he reinterpreted Husayn ibn 'Ali's martyrdom of Karbala' (see below) from a nationalist perspective. Both works represent the cornerstones of the further development of the Bektashiyya. Apart from the Frashëri brothers, several other Bektashis participated directly in the nationalist struggle, by supporting the *çeta* (the resistance bands) or by advocating the diffusion of Albanian writing within the Bektashi *tekke*,[5] the development of which constituted a driving force in the cultural and identity building of Albanian nationalism.

In the newly founded Albanian state in 1913, the Bektashiyya was recognised as an independent religious community. The Bektashi sought to gain this status on their own, establishing their independence from the Hacibektash mother-*tekke* in Turkey, which in the meantime had been closed down by Mustafa Kemal Atatürk's banishment decree. In 1929, as King Zog sought to rationalise the administrative apparatus and institutionalise relations between politics and religion, the Bektashiyya officially became de facto an autonomously structured religious community, namely "Bektashism". Various congresses[6] and subsequent statutes formalised both the nationalist vocation and the essential independence of the Bektashiyya, which now became Bektashism (Clayer 1995). Accreditation in the Albanian nation-state, however, did not prevent the Bektashis themselves from being hit hard by the Communist regime of Enver Hoxha (d. 1985) that took power after World War

II. The Communist regime planned and implemented a gradual atheistic and secularising campaign that aimed to reduce, if not eliminate, the scope of religion in society (Tönnes 1982; Karataş 2020). Several Bektashi *tekke* were destroyed, just as many Bektashis were forced to emigrate elsewhere. One of these was Baba Rexhepi (d. 1995), who led the Bektashi *tekke* in Taylor, Michigan, in the United States, which until the end of Communism was the most representative Albanian Bektashi centre of the time (Trix 2011).

## 3. Bektashi Revival

In 1990, shortly before the collapse of the Communist regime in 1991, religious freedom was restored in an attempt to mark a democratic opening, a freedom that was then seen as a symbol of a break from the past regime, which was considered totalitarian and unjustly atheistic. Nearly half a century of Communist rule nevertheless profoundly transformed Albanian society: the reforms of industrialisation, urbanisation, and mass schooling, while remaining incomplete, had changed the habits of Albanians (Dalakoglou 2012; Lelaj 2012). This period marked the violent transition from a totalitarian, autarkic and statist system to a democratic and open system. Various economically recessive phases followed one another, as did deep political and institutional crises. Occasional social tensions surfaced, exacerbating clan and territorial views that the regime had failed to reset, resulting in a near civil war in the late 1990s. Openness to foreign countries allowed confrontation with new cultural models and led to large-scale emigration as Albanians went to work or study abroad.

Within this landscape, religious freedom was reintroduced to enshrine a break with the Communist past, perceived as pagan and oppressive. Freedom became synonymous with change, and change itself was seen as a source of legitimacy. Yet, anti-religious propaganda and positivism in school teaching, in addition to harsh securitarian measures, had de-educated Albanians about religion, while the new generations had essentially grown up in an atheistic environment, knowing almost nothing about religion. In such new conditions, a complex religious revival ensued (Clayer 2003). In the aftermath of Communism, the population had awakened yearning for religiosity, although few still knew what a ritual or prayer really was. This did not prevent masses of the faithful from rediscovering places of pilgrimage and places of worship where they could express and fulfil, albeit rudimentarily, their religious needs. However, this revival was somewhat de-institutionalised and extemporaneous, i.e., without the religious authorities being able to co-opt it: places of worship were reopened and rebuilt independently, while holy people resurfaced among the population dispensing blessings and prayers toward Albanians in need of reassurance and support for themselves or for their sons emigrating abroad (Bria 2019a, 2019b).

This rebirth from below was matched by a reorganisation from above by the traditional Islamic authorities, who sought to rebuild the religious cult. The Sunni Islamic community has gradually tried to present itself as the bearer of a moderate, nationalist and tolerant Islam (Endresen 2015)—an Islam that is national and state-oriented, capable of dialogue with the government and with other religions without questioning the secularity of the state. This, therefore, requires that such Islam be organised in institutions that are monitored and controlled by the state. It is the so-called "churchification" of Islam, or rather the organisation of organised clerical institutions under state control, that proposes an acceptable religious narrative. According to Račius (2020), the term "churchification" means that the organisation of Islamic communities is domesticated in order to be easily managed and controlled by governments.

Foreign actors have also been involved in religious reconstruction, such as Saudi Arabia, which since the early 1990s has exported its version of Islam by supporting the reconstruction of mosques and offering scholarships to young Albanians to study in Middle Eastern Islamic institutions. Various Turkish foundations have also been active since the 2000s in the field of education: Albanian madrasas are run by Gülen foundations, as is the Beder University in Tirana, where they also teach Islamic studies.

The Bektashi reconstruction followed the same pattern, although it took on its own characteristics that make it a particularly significant case. First, Bektashism reorganised itself around the memory of those of its leaders who were still alive. Many of them returned from abroad to Albania, committing themselves to the front line of the reconstruction of the traditional cult. Bektashi places of worship were reopened as *tekke* and *türbe* ("Islamic tombs"). The latter were often reopened by the descendant families of the Bektashi who had previously run them, although the Bektashi community later took them back under its own management. The great Bektashi pilgrimages were also re-established, first and foremost to Mount Tomor or to Krüja in central Albania, where thousands of believers go to pray to 'Ali 'Abbas[7] or Sarı Saltık[8] (Mentor 2015; Clayer 2017). This reconstruction also involved the revival of the charisma of the Bektashis, whom groups of the Albanian faithful would visit to ask for blessings and healing (Clayer 2007). The *baba* (Bektashi spiritual leaders) resumed performing practices of blessing and healing, making amulets to answer the requests for holiness and blessings by Albanian women seeking support and assistance for their children. Bektashi iconography was reintroduced in a new fashion related to the industrial production and proselytising policies of Iran and Turkish Alevites who imported Shi'i or Alevi icons to Albania (Bria and Mayerà 2017). These icons depicting the first Shi'i imam 'Ali ibn Abi Talib (d. 661), Hajji Bektash and various members of the Prophet's Family, including the Shi'i twelve imams, soon spread to all Bektashi and other Sufi places of worship. The Bektashi sold these icons in their cloisters or gave them as gifts to worshippers; these thus ended up entering the homes of Albanians. Both Muslims and Christians still venerate these icons, which thus became one of the main means of expression of post-socialist religiosity.

Alongside this traditional religiosity, the Bektashi proposed a progressive, nationalist and vaguely New Age religiosity (Clayer 2006). This doctrinal refashioning stemmed mainly from the pursuit of legitimacy in Albania's post-Communist society. From the collapse of the Communist regime until the late 1990s, the Bektashi reorganised their community, partly following in the mould of the pre-Communist period, partly underscoring its self-sufficient and autocephalous tendencies. This reorganisation aimed to grasp community legitimacy in Albania's post-socialist society, in which critical and/or episodic relationships with religion emerged. Influenced by Communist secularisation and the multi-religious discourse of nationalism, Albanians perceived Islam as a domestic but potentially dangerous religion. The aftermath of 9/11 and the de-Islamisation of Albanian politics in the late 1990s and early 2000s shaped a new Albanian Islamic identity in search of a moderate and democratic dimension, which is compatible with Euro-Atlantic values and EU membership. The Bektashi thus roughly followed this trend, reorganising their community in a clerical manner and seeking state recognition, which they then obtained in 2009. On the other hand, they have also tended to present themselves as something different from Islam, as well as from Christianity, thus as a religious third way, that is, an independent and autonomous religious community. This ambition was supported by various Bektashi congresses, which sanctioned the Bektashi doctrinal refashioning.

The statute published in 2000, following the seventh congress of the community, outlined a progressive, humanist, ecumenical and nationalist Bektashism, which would represent a third way between Christianity and Islam, dedicated to the application of the *shari'a* and the spread of tolerant Islamic mysticism. The leaders emphasised the tolerant character of Bektashism, putting local-national Islam above foreign Islam, which was considered fundamentalist. In this sense, Bektashism was labelled as a suitable alternative to Islam, closer to the "occidental" way of life, and featuring democracy, freedom and liberalism. On the other hand, this doctrinal updating corresponded to an organisational rationalisation of the communities, which in this way sought to assume a stable hierarchical structure. Traditional Bektashi doctrinal elements were reaffirmed, such as the worship of 'Ali, the veneration of the Family of the Prophet, and the celebrations of *nevruz* and '*ashura*' (see note 3 below), albeit prevalently reinterpreted in a rather nationalist and progressive key. Bektashism was labelled as a true nationalist cult, thus strengthening the bond that

had existed since the birth of the Albanian state. Albanian flags and banners were often placed inside the Bektashi *tekke* and also displayed during the most important festivities of the year, during which the Bektashi leaders often emphasised their love for the nation. Eminent patriots were sanctified, or rather "Bektashised", such as Naim Frashëri, named "Baba of Honour" and father of the Albanian nation, whilst Bektashi leaders, such as Baba Ali Tomor (d. 1948), were labelled as ardent patriots, or others, such as Baba Martanesh (d. 1947), were regarded as martyrs of the nation.

This also entailed a shift in writing style to one closer to a political record than anything else: messages such as the ecumenism of the nation, peaceful coexistence and interfaith brotherhood in the name of common national belonging became an official Bektashi speech pattern. To the nationalist and ecumenical writing has been added another rather rationalist and scientific one that is often evoked during festivities such as *nevruz* (Bria 2020) or even during the academic symposia that the community organises. *Nevruz*, for instance, which traditionally commemorates the anniversary of 'Ali's birth, is also seen as the advent of spring, of the biological rebirth of nature. National and even international academics are invited to congresses at which topics related to Bektashism are discussed in order to sublimate official discourses, such as the historical origins of Bektashism or interfaith tolerance and the Bektashi. These are events where religious writing is mixed with scientific writing, attended by university professors, intellectuals and Bektashi *baba*s. At some of these events, spiritual rituals are also organised to close the event. The aim is to legitimise the community in the eyes of Albanian society, using a scientific writing style to which they may be more receptive. The reason for this is mainly the socialist heritage and the enduring existence of educational curricula in which a secular, if not positivist, outlook prevails.

This narrative, however, also aimed to emphasise certain traits of the Bektashi tradition, such as pantheism, metempsychosis and the rhetoric of martyrdom, which seemed to be of interest to the Albanians. These discourses also aimed to evoke a kind of primordial, almost pagan religiosity that all Albanians would share, and of which the Bektashi would be the expression. This kind of discourse, however, which would seem counter-intuitively intellectual, is quite common among Albanians due to the Bektashis' ability to disseminate it through various media. Internet and virtual social media are often used to present the image of Bektashism as a nationalist, progressive and ecumenical cult. *Urtësia* ("Wisdom"), the bi-monthly newsletter published by the community, communicates the main activities of the Bektashis, as well as their various anniversaries. However, the Bektashis also organise publications to spread awareness of their history, doctrines and practices between the scientific and the popular. Some of these works are actual translations into Albanian scientific works by foreign scholars, such as Birge's book *The Bektashi Order of Dervishes* or the works of Nathalie Clayer or Robert Elsie on the Bektashiyya. They organised the publication of texts attributed to Hajji Bektash, such as the *Makalat*.

## 4. Neo-Bektashism

The Bektashi community has also promoted works that aimed to be highly representative of Bektashism in the post-socialist era. Among these, the most famous one is *Syri i Tretë* ("The Third Eye") by Moikom Zeqo. The author proposes a singular apologia for Bektashism, describing it as universal ecumenism: a global equilibrium between Buddhism, Christianity, Hinduism, Bektashi Pantheism, Shi'ism and Sunni Islam.

Zeqo was born in 1949 in the city of Durrës in Albania into a Libohova family. After completing his primary and secondary education in that city, he started attending the Faculty of History and Philology in 1967 and graduated from the Faculty of Philology and Albanian Writing in the branch of Writing and Literature in 1971. In the years between 1971 and 1974, he worked as a journalist and as literary editor of the newspaper *Drita*. In the 1970s, he wrote a collection of poems entitled *Meduza*, which broke with the aesthetics prescribed by the Albanian League of Writers and Artists (*Lidhja e Shkrimtarëve dhe e Artistëve*)[9] and criticised Albania's bureaucracy and cultural isolation. When Zeqo attempted to publish poetry from *Meduza*, his work was denounced, and he was stripped of his leadership

of *Drita* and forced to work as a teacher in a small mountain village, Rrogozhina, from 1974–1976. It demonstrated how Zeqo was not really aligned with regime positions. During the years 1979–1987, after four years of "exile" in the mountains, Zeqo was employed by the Durrës Archaeological Museum. Having been a competitive swimmer in his youth, Zeqo turned into a specialist in underwater archaeology, which features prominently in the poem "Zodiac: 2". During this period, he also started publishing "safer" poems and numerous monographs and children's books.

In June 1991, Zeqo was part of the organising group of the 10th Congress of the Socialist Party. In the same year, he was appointed chairman of the Culture Commission and then was appointed Minister of Culture, Youth and Sport. In 1991 he also ran for Socialist Party and again in 1992, when he was elected and served until 1996 as a parliamentarian. Zeqo's active engagement in politics ended with his parliamentary experience, although he did not stop dealing with the affairs of the "nation" indirectly through his publications, in which he let his patriotism be expressed. In 1997, his wife Lida Miraj—also an archaeologist—held a fellowship at Dumbarton Oaks in Washington, DC, which is what brought the Zeqo family to the United States. After returning to Albania, he was the director of the National Museum from 1998 to 2005. From 2005 onwards, Zeqo devoted himself completely to intellectual production, resulting in one of Albania's most prolific and active writers. He was the author of 62 books of poetry, archaeological studies, art history, and numerous screenplays for film and television films of an archaeological and cultural nature.

Zeqo's literary output was eclectic, ranging from non-fiction to fiction and from poetry to more science-based works. One of the common traits of these works, however, was Albanian culture, various aspects of which he addressed, many of them quite sensitive for Albanian scholars. He dealt with the history of Albania, whether in his work on the famous Albanian leader Skanderbeg or the Albanian flag or the culture of the Arbëreshë (Italo-Albanians). He was a profound advocate of the Albanian brotherhood, called Albanianism, which united all Albanian peoples under a single, ethno-linguistic membership that was independent of religion. His publications also included various poems of him, often assembled in collections. Some of his works, however, were related to Hellenic, Babylonian or Egyptian culture, which he considered ancestral, but which were united with Albanian culture. The treatment of these cultures was permeated by a scientific but also popular approach in which Greek or Egyptian myths were mixed with astrology, which he dabbled in. This approach could be called typically New Age or, to some extent, traditionalist, traceable to the Western esoteric tradition, of which Zeqo, however, makes no mention.

*Syri i Tretë*, in some ways, represents a unicum within Zeqo's literary production, as it is the only one that deals directly with a religious tradition, i.e., Bektashism. In other ways, the same New Age approach he uses in other works emerges from this book, as well as the patriotic and nationalist outlook that characterises his various works. He probably decided to deal with Bektashism for two reasons. The first is personal, since his family was Bektashi, as he himself states in the book. Secondly, the Bektashi community involved Zeqo in this work, who was one of the most famous and in vogue writers at the time, in order to enhance Bektashi popularity. Although it is unique in Zeqo's literary production, this book is not the only one to propose a New Age treatment of Bektashi, nor is it the only expression of literary neo-Bektashism or the New Age strand *tout court* in Albania. In fact, there are several authors dealing with occultism, ufology, Greek Egyptian and Albanian myths, many of which have produced works similar to Zeqo's.

One of these authors, Xhevahir Dedej, draws inspiration from Bektashism in his book *Sekretet e Shpirtit* ("The Secrets of the Spirit"), stating that its purpose is to initiate readers into the mysteries of the universe. A pillar of his worldview is that the human soul consists of energy, and his mysterious insights include the revelation that humanity has failed because it has divided religion into different categories (Dedej 2014, p. 45). He states that "Albania is a country where angels have always lived" (Dedej 2014, p. 245) and believes that the Albanians are the chosen people and that he himself is an elected prophet who has inherited his prophecy through a chain of imams.

Another example is Përparim Zaimi's book *Zgjidhja e një misteri* ("The Solution of a Mystery"). For Zaimi, Bektashism in its Albanian form is the ancient religion itself, primordial and universal, and a sign of the existence of other dimensions (Zaimi 2011, pp. 15–17). True Islam is thus pure mysticism (Bektashi), compatible with ufology, paganism and pantheism. Zaimi's desire is to reveal the hidden truth about how humanity and science evolved. He sees the universe as a mysterious and magical whole, in which everything is connected to other dimensions through energy transfers (Zaimi 2011, p. 19). Dedej's and Zaimi's works share with Zeqo's the same New Age, typically post-secular approach, as well as the idea of considering Albanian culture and people as elected ones. On the other hand, Zeqo's work differs in his knowledge of Bektashism and in his ability to deal with other spiritual and religious traditions, such as Christianity, Hinduism, Buddhism, and Greek and Egyptian myth, with the same degree of insight. Zeqo's book can be placed within this strand, although it was probably the most successful and famous one.

Published in 2001, *Syri i Tretë* consists of over 350 pages of essays and poems. Zeqo's writing is very sophisticated and polished, attempting in some sections to demonstrate his mastery of Albanian writing through linguistic virtuosity. He seeks to prove his erudition by making extensive references to scientific texts. In a manner that is not always consistent, he often quotes works by Western scholars, such as anthropologists, Islamologists and historians, for example, Henry Corbin, Annemarie Schimmel, and Alexandre Popovic. He does not always draw inspiration from these books but rather uses them as a tool to demonstrate his erudition as a scholar of international calibre. The book, however, mixes a kind of academic writing with an esoteric and mystical one. The last part of the book contains Vedic texts, Avesta, Buddhist poems (attributed to Siddhartha), Biblical passages, poems by Firdeusi (Abu al-Qasim Ferdowsi), Rumi and Yunus Emre, but also Chinese texts by Lao Tsu and poems composed by Albanian Sufis, including Naim Frashëri. These elements prefigure the approach of Zeqo's book, in which the third eye is that of hidden wisdom, hence esoteric knowledge. This third eye is an esoteric eye that sees beyond the exteriority of the world. An eye that belongs to elected humans, such as the ancient Egyptians, the Hindus, and Shiva, but also to Bektashism, which represents the third eye between Christianity and Islam. Bektashism is, hence, the starting point and source of inspiration for Zeqo, who reworks its doctrines to bridge with other mystical traditions—Hindu, Buddhist, Confucian—and also psychology, alchemy and the hard sciences.

## 5. New Age and Bektashism

The centrepieces of Zeqo's discourse, however, are both Bektashism and Naim Frashëri, who would be the exemplary man, obviously endowed with the third eye of knowledge, who would instil Bektashism, and through it all Albanians, with the ability to possess the "chosen knowledge". Through *Syri i Tretë*, which stands between the scientific and the esoteric, Zeqo attempts to argue for the idea that Albanians grasp the esoteric (hidden) third eye of human knowledge, using various narrative levels or registers to which correspond recurring discourses of Bektashism or of more typically New Age and occultist religions. Zeqo's writing is also political due to its marked nationalistic traits, which the author emphasises at various stages in order to name the Albanians as a chosen people, thus ontologically superior to all others, and the origin of this superiority would be precisely in writing, as the chosen method of transmission of knowledge to all Albanians. It is no coincidence that Zeqo devotes special attention to the semantics of Albanian words and proposes an analysis of every single letter of the Albanian alphabet from which a branch of wisdom would originate (Zeqo 2001, p. 45).

Zeqo's book starts with Naim Frashëri, who is described as a "miracle of the Albanian intelligentsia of all times", and a "master of modern Albanian literature" (Zeqo 2001, p. 14).[10] His life is regarded as an apostolic work, or rather "as a Christianity without Christ" (Zeqo 2001, p. 16). Zeqo considers him a great patriot, equating him to the Albanian leader Skanderbeg, but at the same time, alludes to his almost divine nature in his apostolic

work. Zeqo also writes that "he was a poet who built the prophecy of the future" (Zeqo 2001, p. 15) and believes that no one would understand both the national and universal awareness of humanity like him.

According to Zeqo, Naim Frashëri was an outstanding missionary of both national and world Bektashism, but he was first and foremost an initiator of religious ecumenism, which would result from the "fusion of all religions into a single anthropological, monotheistic, philosophical and mystical model". Frashëri redrew the boundaries of conventional global politics, disregarding any ethnic individualism. He was thus not only a very important intellectual of the so-called Albanian renaissance (*Rilindja*) but a global and modern intellectual, a bearer of the Socratic teachings, for which he belonged to all mankind. His books, written in Greek, Turkish and Persian, attest to his polyglot and global outlook. He was, therefore, an expression of the global enlightenment, which connected all enlightened people like him (Zeqo 2001, p. 23).

The book incenses the figure of Naim Frashëri, a true universal role model, a learned, cultured and at the same time divinely inspired man. This model is quite like that of Ibn 'Arabi's perfect man, a doctrine quite widespread and rooted in Bektashism, as well as in other Islamic mystical traditions (Morrissey 2020). The figure of Frashëri, as outlined in the book, seems to trace these overall qualities of his both in his human and mystical works. Indeed, his belonging transcends Islamic boundaries to be something more global and all-encompassing, belonging to the whole of humanity. However, the author does not totally universalise Frashëri, as he always claims a national and religious affiliation with Bektashism.

These traits are then elucidated in the rest of the book, where the author focuses on rather scattered references to Bektashism and its connection to history, philosophy, science, Islam and other religious traditions. Zeqo mentions in the book that his father was a Bektashi. In fact, he shows that he is well acquainted with the Bektashi doctrines and the history of the order. Zeqo identifies various sources of Bektashism, including the Qur'an, which is said to have left an indelible mark on Bektashi philosophy. Another source would be the martyrdom of Husayn ibn 'Ali in Karbala', which he equates with the martyrology of Christ, positing a syllogism between Christianity and Bektashism. Another source would be the first imam, 'Ali, who would be the basis of Bektashi theology. This would imply four modes of understanding: exoteric understanding (*zahir*), esoteric (*batin*), moderation (*hadd*) and divine vision (*muttala'*). He also cites the well-known French historian Henry Corbin as a source for his work.

For Zeqo, Bektashism would imply an esoteric approach to Islamic knowledge, which would prescind a legalistic or literal approach (Zeqo 2001, p. 45). Bektashism thus implies a gnostic and mystical knowledge of the Qur'an. Although rooted in prophetic revelation, Bektashism would be "a prophetic philosophy, one of the branches of Persian Shi'ism. Bektashism would thus encompass Shi'i imamology, i.e. a mystical philosophy that would complement Islamic prophetology" (Zeqo 2001, p. 125). Zeqo also dedicates a section of his book to Hajji Bektash, regarded as an exceptional mystic who excelled in poetry and miracles: "Hajji Bektash have distinguished himself in the enlightenment of the eastern Islamic world, a reformer such as St. Francis of Assisi who would reform Christianity" (Zeqo 2001, p. 156). He would reform the Islamic religion and formulate the Bektashi doctrines.

At the same time, it is evident that there is a New Age or rather perennialist[11]/occultist[12] vocation in Zeqo's work, which tends to establish a communion between all religious traditions in history, deriving a single ontological matrix from them. This is also the case with Naim Frashëri, who would thus be a chosen one who inherited a kind of enlightened wisdom that would be handed down in history through various characters. Socrates, Osiris, Aristotle, St. Augustine, 'Ali and even the sixth Shi'i imam, Ja'far al-Sadiq, would be characters who received this enlightenment, culminating in Naim Frashëri, who then condensed it into the "Naimian message" (Zeqo 2001, p. 148). Inherent in this idea is the existence of "chosen" characters who are the only ones capable of grasping this

enlightened knowledge. At the same time, there are chosen peoples who embody the enlightened knowledge of humanity. These peoples include the Albanians, who, through Naim Frashëri, received the enlightenment to be the vanguard of humanity. The "Naimian message" is thus the perfection of this hidden knowledge handed down over the centuries through various characters and various peoples, but which finds its zenith in Naim, the Albanian Bektashi.

The Bektashi doctrines, like its history and that of Hajji Bektash, are thus re-read in a typically New Age or perhaps occultist key. Zeqo is adept with his knowledge at creating a bricolage in which Bektashi elements are mixed with other religious and spiritual traditions, such as Christianity or Hinduism, but also the cult of the ancient Egyptians or Greeks. All elements that Zeqo knows very well and does not hesitate to use to propose Bektashism as a global religion, but at the same time as a typical Albanian one. A tension between universalism and particularism emerges, which will be analysed in the next section, but which at the same time is rooted in typical post-secular religiosity.

## 6. Between Particularism and Universalism

Zeqo's book reveals a tension between a universalist and a particularist conception of Bektashism. The former envisages Bekashism as a universal tradition common to all mankind; the latter, on the other hand, tends to consider Bektashism in its connection with Albanian culture as elected and, therefore, the only one capable of grasping esoteric divine enlightenment. This tension is rooted in Zeqo's deep personal convictions, but at the same time in a political vision—typically patriotic and nationalist—that considers Albanian culture as "special", in which, however, an overall ecumenism according to which all religious traditions are connected and united is present.

It would be Naim Frashëri who definitively reformed Bektashism, mixing it with Albanianism, to make it "a universal world philosophy, capable of representing a moral and ethical guide for the community" (Zeqo 2001, p. 87). A guide that also provided support to Albanian politics, in difficulty given the time of change, by providing guiding and reference values. Values that were set out by Frashëri in his work *Qerbelaja*, in which he outlines an all-inclusive ecumenical structure, a structure that would be based on an unquestioned monotheism. For men belonging to various religions, there would be only one anthropological truth: we all come from the same earth. Naim's ecumenism is the core of Bektashism, being embraced by all Albanians. Therefore, "Bektashian ecumenism is the ecumenism of all Albanians, which is the conceptual platform for world ecumenism" (Zeqo 2001, p. 67). For Zeqo, world ecumenism is a global political theory and a biological and political anthropomorphism (Zeqo 2001, p. 67).

Zeqo, therefore, reworks the Bektashi doctrines to propose a universal and inclusive model capable of involving other religions as well. This is obviously a legacy connected to the so-called Albanian national religion, according to which all religions are equal, so there is only one God who unites all human beings. At the same time, it is an adage of various traditionalist and occultist doctrines, according to which there is one and only one unifying ontological matrix. However, Zeqo argues it through Bektashism, which is elevated to a worldwide ecumenical model, ergo to be imitated. This is yet a discourse that encounters a tendency on the part of the Albanian Bektashi authorities to pose as a world unicum, that is, an ecumenical and inclusive religious cult that disregards any form of radicalism and division. A discourse that obviously endorses nationalist rhetoric but equally appeals to the Albanian political authorities, who seek in various ways to present the Albanian multi-faith state as a model to be exported.

From Zeqo's book, however, emerges a political, more properly particularistic nationalist register that tends to represent the background of his work. This yearning is mixed with and finds its fulfilment in the perennial doctrine of the chosen people. Albania and the Albanians are regarded as people who are almost privileged to have a certain election. This would be revealed by Naim Frashëri, who, through his reworking of Bektashism, would awaken the vocation of the Albanian people. Bektashism, in fact, finds its fulfilment in the

Albanian people, while at the same time, Albanianism was awakened by Bektashism. This dialectical relationship would find its fulfilment in Naim Frashëri, who set "the course for the development of mankind in his famous work, *Qerbelaja*" ([Zeqo 2001](), p. 105).

Zeqo compares the tragedy of Karbala', in which Husayn in 'Ali, grandson of the Prophet Muhammad and son of 'Ali, was defeated and killed, to the great battles that have marked humanity, such as the battle of Troy. Being the main subject of the work *Qerbelaja*, Karbala' is also comparable, according to the author, to Skanderbeg's struggle against the Turks, when he historically resisted in his stronghold ([Zeqo 2001](), p. 106). Shi'i martyrology is thus mixed with nationalist rhetoric, as when Zeqo compares Karbala' to the 1999 Kosovo war. The latter war has remained etched in the memory of the Albanians, such that the rhetoric of a greater united Albania, in which the Albanian peoples of Albania, Kosovo, Macedonia and Montenegro were included, is back in vogue. For Zeqo, Frashëri respected the basic structure of the Karbala' tragedy to propose a model of universal piety that could symbolise all the abuses and injustices of human history. Here too, it is a universalisation of a concept, that of Shi'i martyrology, to the whole world. This universalism becomes particularism when it refers to the Albanian people and to their writing as an expression of Albanian chosenness. For Zeqo, Naim Frashëri had elevated the literary and poetic model of Albanian writing through his verses, which were comparable to those of other famous poets such as Homer or William Blake. However, Naim is the apogee of a long tradition of extraordinary Albanian poets, such as Girolamo de Rada (d. 1903), considered one of the fathers of Albanian writing.

### 7. Conclusions: Post-Secular Religiosity?

> What is the Third Eye? The third eye is that of Shiva. The third eye is that of Polyphemus. The third eye is Naim Frashëri himself, between the eye of earlier ages and the eye of later ages. The third eye is Albania, between the eye of the East and the eye of Naim's creativity. The third eye is Shqipëria between the eye of the East and the eye of the West. ([Zeqo 2001](), p. 221)

This passage summarises Zeqo's thoughts on Bektashism and Naim Frashëri. Both are a synthesis of various traditions and eras. References to the Indian or Greek tradition, however, indicate the presence of a tradition, a *fil rouge* in history, which found its culmination in Naimian Bektashism. Bektashi doctrines thus have mingled with Albanian culture, creating a universal and elected model, a third way, between West and East, between antiquity and innovation, and between science and religion. The third eye is thus an elected eye, but also a third way, going beyond Christianity and Islam, to propose another inclusive and ecumenical model.

Moikom Zeqo's work appears to be a New Age reworking of the Bektashi doctrines, as is the case with other mystical traditions—not only Sufis. The New Age reinterpretation of Sufism is, in fact, widely recognised as a rather widespread phenomenon in Europe and the US that seems to have some relevance elsewhere as well ([Sedgwick and Piraino 2021]()). There is no evidence of the direct influence of Western esoteric or New Age doctrines in Zeqo's book, which nevertheless does not exclude them. Perhaps he was able to acquire some sources through unconventional channels, or he just did not mention them. However, it would be reductive to flatten this work to a mere intellectual and New Age elaboration of Bektashism, as it is highly representative of Bektashi reconstruction in the post-socialist era. This work is influenced by the immense literature the author consulted, which mostly concerned works by scholars of Sufism and Balkan history. In his bibliography, he mentions Irène Mélikoff, Alexandre Popovic, Thierry Zarcone, Henry Corbin or Annemarie Schimmel. The citation of these works serves mainly to give an almost scientific value to Zeqo's book, which aims to be a compendium of doctrine, history and poetry. His writing lies, in fact, between positivism and mysticism, mixing science, mysticism and poetry, a mixture that can be considered one of the results of radical socialist secularisation.

The Albanians were, in fact, socialised within a basically secular social context, in which the educational systems still largely have a positivist approach. Religious writing

would therefore be unfamiliar to most. Zeqo's work, consciously or unconsciously, embodies a secularist engulfment of post-socialist or even post-secular religion, of which Bektashism is a direct expression. Therefore, his work is not marginal but reveals a common trend in Albanian society that is not only common to the members of the Bektashi community. In fact, many Albanians—without necessarily being Bektashis—consider Bektashism to be a sect in which it is possible to detect an ancestral, almost pagan, religiosity that is common to all Albanians. This idea would thus tie in with the discourse of the national civil religion, according to which there exists an ontological bond between all Albanians, disregarding any other religious tradition. At the same time, Bektashism is a typically post-secular phenomenon. Anti-religious struggle, positivist propaganda and social and institutional secularisation have de-institutionalised religion, demineralising it of all formal aspects, and it ends up remaining a vague reference to the sacred, actually without any particular reference. Albanians mainly prayed, secretly during Communism and openly afterwards, to a sacred object or relic without thinking about what the reference tradition was, thinking that all religions were equal. An attitude that is the sublimation of nationalist discourse, but on the other hand, it is the result of a diseducation of religion and disintegration of religious memory, so decisive in delineating religious practice and even belief. For the Albanians, religion is a post-secular fact, implying a religiosity profoundly changed by secularisation (and not a denial of secularism). Optionality, flexibility and ontological security searching are among the main characteristics of post-secular religion, which also implies the emergence of new spiritualities and/or religious solutions. The latter may be individual as well as collective, but they certainly intercept a common feeling that characterises late or post-modernity, such as Zeqo's work. Although *Syri i Tretë* is a sui generis work, it expresses typically post-secular religiosity, in which Bektashism is reinterpreted in a New Age key to propose a spiritual solution or at least a spiritual narrative that neither the Albanians nor the leaders of the Bektashi community disapprove of, but rather legitimise.

The references to 'Ali, the Shi'i twelve imams, or the martyrology of Karbala' are well present in Zeqo's work but reinterpreted between the particular and the universal. Particular in that they are connected to Albanian history, as in the case of the Kosovo war being compared to Karbala'; general, in that the martyrdom itself is considered a global event. This tension—between the particular and the universal—is constant throughout the book: Bektashism's quest to be a universal model, somewhere between Christianity and Islam, a third tradition that can unite all of humanity; Bektashism's quest to be an elected model in its union with Albanian culture, to make the Albanians an elected people. A tension that Zeqo also builds using the tools of occultism, which, however, seems to derive more from the evolution of Bektashian thought in the post-socialist and post-secular era than from an external takeover.

**Funding:** This research received no external funding.

**Institutional Review Board Statement:** Not applicable.

**Informed Consent Statement:** Informed consent was obtained from all subjects involved in the study.

**Conflicts of Interest:** The author declares no conflict of interest.

## Notes

[1]  For a broader study of the relationship between "mysticism" and "modernity" and various aspects of their reciprocity, entanglement and harmony, see Zarrabi-Zadeh (2020).

[2]  Janissary, also spelled Janizary, Turkish Yeniçeri ("new soldier" or "new troop"), signifies a member of an elite corps in the standing army of the Ottoman Empire from the late 14th century to 1826. Highly respected for their military prowess in the 15th and 16th centuries, the Janissaries became a powerful political force within the Ottoman state.

[3]  The Bektashi doctrines attest to a certain centrality to the veneration of the first Shi'i imam 'Ali, but also in general to that of the Prophet's Family (*Ahl al-Bayt*), including the Shi'i twelve imams. Such veneration assumed some distinctive hallmarks, such as various divine attributions assigned to 'Ali, in a manner close to some extreme Shi'i sects (Asghari 2008). The numerology

and science of letters of Hurufism was also present, especially in the Bektashi iconographic representations ([Birge 1937](#)). From a ritual point of view, the Bektashis observe fasting during the first days of Muharram, the first month of the Islamic calendar, in commemoration of the Karbala' massacre, and especially the tenth day of Muharram, which marks the martyrdom of Husayn ibn 'Ali. *Nevruz*, as the anniversary of 'Ali's birth, is also celebrated. In any case, the epic recital and mourning associated with the Karbala' massacre marked Bektashi piety, as evident in practices of ritual weeping or corporal mortification (chest blows), or in the literature of poets attributed to Bektashi milieus, such as Yunus Emre (d. 1321).

4    Kadızadelis (also Qadizadali) was a seventeenth century puritanical reformist religious movement in the Ottoman Empire, whose members followed the revivalist Islamic preacher Kadızade Mehmed (d. 1635).

5    The *tekke* is an architectural structure erected for a Sufi order and is a place of spiritual retreat.

6    Since the end of Communism, congresses of Albanian Bektashism have been held in 1993, 2000, 2005 and 2009. They redefined its institutional structure and the Bektashi creed as known today.

7    'Abbas ibn 'Ali (d. 680), the son of 'Ali and Fatima, is highly revered in Shi'i Islam for his loyalty to his brother Husayn ibn 'Ali and for his role in the Battle of Karbala', in which he was the standard-bearer for the *Ahl al-Bayt*.

8    Sarı Saltık (d. ca. 1297) is a 13th-century dervish venerated by the Bektashis.

9    The Albanian League of Writers and Artists is an organisation of writers, composers, and artists and critics of literary and artistic values, located in Tirana. During the Communist period, the League was a tool of the government's efforts to require writers and artists to advance the goals of the Regime and to censor, ban, and punish those writers and artists who failed to do so.

10    All translations from *Syri i Tretë* in this article is from the author.

11    Perennialism is the view that there is a shared core of truth and perennial wisdom in all major religions—sometimes called a perennial philosophy—and that this core is grounded in and justified by shared religious experiences, usually of the mystical variety ([Draper 2020](#)).

12    Occultism is a branch of human activity and is an orientation towards hidden aspects of reality, those that are held to be commonly inaccessible to ordinary senses. Such activity simultaneously shares a certain similarity with both science and religion but cannot be reduced to either of them ([Bogdan and Djurdjevic 2014](#)).

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
