# Peer review of "Nationalism, Post-Secular and Sufism: The Making of Neo-Bektashism by Moikom Zeqo in Post-Socialist Albania"

_religions, doi:10.3390/rel13090828_

Round 1

Reviewer 1 Report

The topic of the article is highly relevant and several dimensions of the analysis are very important for the understanding of evolutions of Sufism at the beginning of the 21st century, in particular in a postsocialist context. These dimensions appears clearly in the very good conclusion : a new-age reworking of the Bektashi doctrine ; the fabric of a post-secular religion ; the existence of a tension between the particular and the universal.

However the body of the article is to be reconsidered and reelaborated, firstly because the first half of the paper is dedicated to the "context" (here the history of Bektashism in the Albanian space) and only 4 and a half pages to the analyses source (the book of Moikom Zeqo), and secondly because the article should offer a deeper critical approach of the work and the ideas of the author in question.

Thus I would suggest to shorten the historical part and to concentrate, even to start from the post-socialist context (which means of course coming back to certain aspects of the socialist period, and even when necessary to other periods, such as the end of the 19th-beginning of the 20th century when the figure of Naim Frashëri is ti be explain). 

The personnality and the positionning of the author should be stressed in a moore analytical way (his clear position within the communist system, his double profile, as a historian and archeologist and a s a poet and writer (and even as a politician). The rest of the production of the author should be sketched, in order to understand the specificity or not specifity of "Syri i tretë" in his own production. The personnality and work of Moikom Zeqo have to be seen throw the dimensions pu forward in the conclusion.

The content of the book should be presented. It seems that the work is very specific, be it in the  literary field, in the field of new-age literature or in Albanian the Bektashi production (by the way it does not seem that the book is "the manifesto of Albanian Bektashism", or could the author of the article give clues for such a statement).

Then, the analysis of the content could be articulated around the three dimensions that the author of the article put forward in the conclusion.

Author Response

Thank you very much for your suggestions, which I have tried to follow to reshape my work. I have reduced the historical part, giving more emphasis to the content of the book. I used the theoretical tripartition of the conclusions: a new-age reworking of the Bektashi doctrine ; the fabric of a post-secular religion ; the existence of a tension between the particular and the universal. I used the category 'the fabric of a post-secular religion' as an interpretative key to the conclusions, which I found very useful. I also provided a broader discussion of the author and his literary production. I agree that the work is a new-age reworking of Bektashism.

Reviewer 2 Report

Dear author(s),

With interest I read your concept paper. It shows that you are very much versed in the subjet under concern. Still, having to be critical, I missed a methodological section in which you make clear to the reader what exactly is your research question and what is your methodology. As it is now the article is more of a review, put in historical and societal context of Albania of the work Syri i Tretë of Mikom Zeqo. And you are doing a good job in that respect because, as I said, it shows that you are an expert or experts in the field. So my only comment would be to transform the article from the review it is now to a more academic text with an old fashioned research question and a justification of how you analyzed the book. I would as well use more quotes from the book (there are quite scarce now) to reinforce your conclusions at the end (mind to quote the page numbers!). Success, the reviewer.

Author Response

Thank you very much for your suggestions, which I have tried to follow to reshape my work. I have reduced the historical part, giving more emphasis to the content of the book. I used the theoretical tripartition of the conclusions: a new-age reworking of the Bektashi doctrine ; the fabric of a post-secular religion ; the existence of a tension between the particular and the universal. I also provided a broader discussion of the author and his literary production. I have also included a section on the methodology adopted in the work.

Round 2

Reviewer 1 Report

.

Author Response

I agreed to the changes suggested by the editor which rendered the article much more fluent and meaningful. 
